# Using Household Dietary Diversity Score and Spatial Analysis to Inform Food Governance in Chile

**DOI:** 10.3390/nu16172937

**Published:** 2024-09-02

**Authors:** Martín del Valle M, Kirsteen Shields, Sofía Boza

**Affiliations:** 1Global Academy of Agriculture and Food Systems, The University of Edinburgh, Edinburgh EH8 9YL, UK; kirsteen.shields@ed.ac.uk; 2Department of Management and Rural Innovation, Faculty of Agricultural Sciences, University of Chile, Santiago 8330111, Chile; sofiaboza@uchile.cl

**Keywords:** Chile, food security, food governance, Household Dietary Diversity Score, spatial visualization

## Abstract

This study explores how the Household Dietary Diversity Score (HDDS) and spatial visualization can inform food governance in Chile, focusing on socio-demographic and geographical determinants affecting food consumption patterns. A national household database (n = 4047), including households from 2019 (n = 3967; 98.02%) and 2020 (n = 80; 1.98%), provided by the “Family Support Program of Food Self-Sufficiency” (FSPFS) of the Ministry of Social Development and Family, was analyzed. The findings revealed that Chilean vulnerable households were led mostly by women (86.6%), with an age average of 55.9 ± 15.6 years old, versus 68.9 ± 12.9 years in the case of men. The intake frequency analysis showed that dairy, fruits, and vegetables were below the recommended values in at least half of the households, and that fats and sugars were above recommended levels. Regarding the HDDS (0–189), the national average was 91.4 ± 20.6 and was significantly influenced by the number of minors in the households, water access, food access issues, and residing in the Zona Sur. Finally, the spatial visualization showed that the Zona Central had higher consumption of fruits and vegetables, while the extreme zones Norte Grande and Zona Austral showed higher intakes of fats and sugars. These findings emphasize the importance of leveraging data insights like the HDDS and spatial visualization to enhance food security and inform food governance strategies.

## 1. Introduction

### 1.1. Household Dietary Diversity as an Indicator of Food Security

Household dietary diversity is commonly defined as a qualitative measure of food consumption at the household level that reflects access to a variety of foods and serves as a proxy for nutrient adequacy (Refs. [1,2,3]) and micronutrient adequacy in resource-poor settings [4], and encompasses a range of methodologies provided by different approaches related to the number of food groups and to whom they are directed. For example, ref. [1] highlights the Household Dietary Diversity Score (HDDS), Child Dietary Diversity Score (CDDS), and Women Dietary Diversity Score (WDDS). Refs. [2,4] introduced the Minimum Dietary Diversity for Women (MDD-W) score, which is based on the consumption of 10 specific food groups. Refs. [5,6] provided the Household Dietary Diversity Score (HDDS), focusing on 12 food groups for assessment. Ref. [7] adapted the MDD-W methodology by using 10 food groups as a basis for their measurement. The Food and Agriculture Organization suggests a dietary diversity questionnaire underlining the importance of considering various food groups to gauge the extent of dietary diversity in individuals or households [3].

The HDDS can be assessed through various indicators and components, each shedding light on the variety and quality of foods consumed within a household. Ref. [8] indicate that these methodologies need to be simple to better predict micronutrient adequacy. According to [1], the Household Dietary Diversity Score (HDDS) is a commonly utilized indicator, calculated based on the number of food groups consumed by the household over specific timeframes, typically within either 24 h or 7 days. This score has proven to be a reliable predictor of nutrient adequacy in the diet and typically consists of 12 food groups, including cereals, vegetables, fruits, meat, poultry, eggs, legumes, milk, oils, sweets, spices, and beverages. Another approach, highlighted by [6], involves assessing the number of different food groups or types of food consumed by household members over a specified period. The Household Dietary Diversity Score (HDDS) is a valuable tool in this regard, ranging from 0 to 15, reflecting the number of food groups consumed. Households are categorized based on their dietary diversity, generally classifying those consuming at least four different food groups (DDS ≥ 4) as having medium dietary diversity. Ref. [7] discusses an adapted version of the Minimum Dietary Diversity for Women (MDD-W) methodology, focusing on 10 defined food groups to assess dietary diversity. Respondents recall their food consumption over a 24-h period, and enumerators categorize the foods into these 10 groups, ultimately assigning a dietary diversity score (DDS) out of 10. Similarly, Ref. [2] use the Minimum Dietary Diversity for Women (MDD-W) score, assessing whether women have consumed foods from five or more of the 10 defined food groups in the previous day. These food groups encompass a wide range, including grains, pulses, dairy, meat, poultry, eggs, and various fruits and vegetables. However, these methodologies also present limitations, as stated by [8], like small sample sizes in some data sets and potential underreporting and overreporting of dietary intakes, or food groups misreported when adding foods in small quantities to sauces [9].

A diverse diet that includes a variety of food groups is generally associated with better nutrient intake and overall health (Refs. [9,10]). These chosen dietary diversity indicators, such as the Household Dietary Diversity Score (HDDS) or Dietary Diversity Score (DDS), are intrinsically linked to food security by quantifying the variety and quality of food consumed. Ref. [10] emphasized the vital role of policies and programs aimed at bolstering household dietary diversity to ensure access to a diverse array of foods, ultimately improving food security. Ref. [11] demonstrated positive correlations between dietary diversity indicators and macro/micronutrient adequacy in various age groups. Strategies like enhancing market access for farm produce and generating off-farm employment are recognized as effective means to boost dietary diversity, thus positively impacting food security. Additionally, dietary diversity indicators serve as invaluable monitoring tools to assess the efficacy of food security interventions. Ref. [5] highlighted the practical utility of the HDDS in identifying vulnerable households requiring targeted food security interventions, particularly during crises like the COVID-19 pandemic. Ref. [6] emphasized the role of their DDS in enhancing dietary diversity, which contributes to improved food security and nutrition while also promoting agricultural sustainability through diverse crop cultivation. Ref. [12] showed in Burkina Faso that higher intakes of organ meat, flesh foods, vitamin A- and vitamin C-rich fruits and vegetables, and legumes and nuts were significantly associated with a lower risk of micronutrient inadequacy. Ref. [7] further underscored the connection between dietary diversity and food security, advocating for the incorporation of dietary diversity into policymaking to enhance nutritional quality and overall well-being.

### 1.2. Spatial Visualization Food Insecurity Warning Systems

Spatial visualization is a crucial tool in addressing food insecurity, providing a geospatial lens to understand its distribution and severity. Refs. [13,14] highlighted the value of spatial visualization in identifying high-risk areas, thereby supporting effective intervention targeting. Similarly, Ref. [15] emphasized the utility of spatial analysis, particularly mapping, in identifying unmet needs, facilitating a comprehensive understanding of the relationship between poverty, population density, and food access. Ref. [16] underscored the practical significance of visualizing food insecurity in prioritizing interventions, as evidenced by their research in Rajasthan, India. These tools enhance the targeting of vulnerable populations, facilitating effective decision-making and resource allocation (Refs. [13,17,18]). By mapping spatial patterns, these tools offer visual representations of food insecurity across regions (Refs. [13,19]), aiding policymakers in targeting interventions and understanding local geographic influences (Refs. [19,20]). Additionally, these tools reveal spatial disparities, informing sustainable territorial-based agriculture and food security policies [19]. They also highlight significant household and individual-level variations, capturing diverse factors influencing food security and dietary quality [18]. Indeed, these tools prevent oversimplification of food insecurity, enhancing the depth and accuracy of information [20]. The integration of geo-referenced data collection via handheld devices has reduced costs, enabling better data collection and targeted interventions in urban areas (Refs. [13,20]). Finally, spatial visualization tools enable comparison of different geographic resolutions, offering new perspectives for policy research and providing a comprehensive view of deprivation and food insecurity [20]. They aggregate summaries of food stores, food banks, and bus stops, contextualizing the entrenched issues of food insecurity.

There are several successful case studies and applications that showcase the effectiveness of different spatial visualization tools and methodologies in addressing food insecurity. For example, Ref. [13] highlighted the Integrated Food Security Phase Classification (IPC) system by the World Food Programme (WFP), a standardized approach employed in over 30 countries. The IPC system classifies food security outcomes at different geographic levels, allowing for cross-country comparisons and targeted interventions. Ref. [17] described a successful web-based prototype developed in Bogota, Colombia, which integrated diverse data sources such as census, socioeconomic, and accessibility data to visualize community kitchens and food resources. This prototype aided in the spatial exploration of food security challenges in specific neighborhoods. Refs. [18,19] emphasized the use of GIS-based indicators and spatially explicit methodologies for mapping local spatial interactions and identifying geographically deprived areas and clusters with high concentrations of food insecurity hotspots. Furthermore, Ref. [19] discussed the integration of spatially targeted interventions to combat inequalities and improve household livelihoods and welfare, particularly in western Kenya. The application of small area estimation (SAE) was highlighted by [13] as a successful method for estimating food insecurity indicators at the district level in Bangladesh, providing precise and representative estimates that are valuable for resource allocation and policy-making. Additionally, Ref. [18] showcased the efficiency and cost-effectiveness of tablet-based data collection, as demonstrated by [21] in conservation projects, achieving cost reductions of up to 75% compared to traditional paper-based surveys. Finally, Ref. [14] conducted a critical review and mapping of indicators to measure the food access dimension of food security, providing valuable insights for assessing and addressing food insecurity.

Despite the advantages highlighted before, spatial visualization also faces important limitations. One of the main restrictions is the limited availability of high-quality subnational data, especially in low-income countries, which can affect the accuracy of spatial analysis [13]. Additionally, there is the potential for misinterpretation of spatial data, particularly by users unfamiliar with data acquisition methods, and the possibility of spatial bias due to non-representative data or poorly defined spatial units. In addition, spatial visualization tools rely heavily on limited data sources and require continuous updates to maintain their accuracy. These tools are primarily based on census and survey data collected at infrequent intervals, which can restrict the timeliness and precision of the information (Refs. [17,20]). The validity and accuracy of data based on self-reported measures by households, and the need for additional validation through local surveys, highlight significant challenges (Refs. [19,20]). Furthermore, the high costs associated with collecting spatially disaggregated data, especially through in-person surveys using electronic devices, further complicate their implementation, and the complex socio-ecological interactions involved in food insecurity require a precision approach tailored to individual and neighborhood-level factors, which can be challenging to execute [18]. There is also a notable bias towards those with access to the platform originating the observed data and significant difficulty in linking heterogeneous data sources to enhance insights [20]. Also, potential differences in food insecurity experiences between urban and rural areas, driven by the clustering of emergency food assistance and supermarkets around population centers, need to be considered [20]. Lastly, the risk of misinterpretation of maps by policymakers and the need for continuous updates to ensure the accuracy and relevance of spatial information pose further limitations [13].

### 1.3. Research Rationale and Objectives

The most recent “Report on Food Consumption in Chile” [22] highlighted a concerning trend: all income quintiles are consuming healthy food groups below the recommended levels, with the lowest income quintile being the most affected. In an average household of 3.3 people, monthly consumption includes 23.4 L of sweetened beverages, 17.5 kg of bread, and 5.1 kg of sweets, whereas the intake of fruits, vegetables, and legumes barely reaches 24.7 kg [22]. Furthermore, the latest National Socioeconomic Characterization Survey [23] indicates that 16.3% of Chilean households face moderate food insecurity, with 3.5% experiencing severe food insecurity, largely due to financial constraints. These statistics underscore the pressing challenges faced by vulnerable households, particularly those in the lower income quintile, in achieving the recommended dietary intake. Understanding the dietary diversity, food security status, and geographical location of these households is crucial, as they encounter the most significant barriers to consuming essential food groups such as fruits, vegetables, and legumes.

This research aims to provide a comprehensive assessment of the dietary habits, socio-demographic characteristics, and geographical distributions of these vulnerable households. The insights gained will be vital for informing strategies and interventions to address food insecurity and enhance the well-being of the most marginalized segments of the Chilean population.

Thus, the main objective of this study is “to analyze the interplay between socio-demographic factors, food consumption patterns, and food security in Chile’s most vulnerable households, and to develop a spatial visualization information system that visually identifies and warns about areas of food insecurity across different food groups at both national and regional levels”. In order to deepen the analysis, the following secondary objectives have been proposed:-To describe the socio-demographic characteristics of Chile’s most vulnerable households, represented by the “Family Support Program of Food Self-Sufficiency (FSPFS)” diagnostic survey.-To calculate the intake frequency of different food groups in Chile’s vulnerable households at the national and macro-zone levels.-To calculate a national and regional Household Dietary Diversity Score (HDDS) and analyze its association with socio-demographic characteristics, food security determinants, and geographical macro-zones.-To develop a geospatial warning system based on spatial visualization of food-insecure areas for different food groups.

## 2. Materials and Methods

The current research received ethical approval from the Human Ethical Review Committee (HERC) of the Royal (Dick) School of Veterinary Studies at the University of Edinburgh, with approval number HERC_2022_148 on 31 October 2022. The approval letter is available in Appendix A, and it was granted on the condition that the research is conducted according to the description provided in the application and the assurances made.

The methodological framework for this chapter was designed to dissect the socio-demographic and nutritional contours of Chile’s most vulnerable households, based on the work done by [5] focused on the determinants of household food security and dietary diversity during the COVID-19 pandemic in Bangladesh. Segregated into four strategic objectives, this chapter evaluates household composition, dietary intake patterns, the HDDS, and the development of a Spatial Warning System, each section providing a critical lens on different facets of food security. For the purposes of this research, the HDDS methodology was chosen due to its generality over other, more specific methods of measuring dietary diversity in households, such as methodologies focused on women suggested by [1,2], or on children [1]. For the spatial visualization of food security, Ref. [13] was used as a reference; however, spatial clustering analysis was not conducted due to concerns about low representativeness and high sensitivity to small data variations, which could hinder the identification of significant spatial patterns.

Households’ information was gathered from a database that contains the results of the diagnostic survey for the Family Support Program of Food Self-Sufficiency (FSPFS) (Programa de apoyo a familias para el autoconsumo) of the Ministry of Social Development and Family of Chile and the Solidary Fund of Social Innovation (FOSIS), which was obtained through a request for transparency and confidentiality agreement to the Secretary of Social Services of said ministry. The households invited to participate in this program are those belonging to the 40% most vulnerable population in Chile, according to the socio-economic characterization survey conducted nationwide. This program seeks to increase the availability of healthy food for vulnerable families, through education and self-provision to supplement their food needs and improve their living conditions. It mainly considers support for production activities (cultivation and breeding of small livestock) and, secondarily, activities aimed at the preservation, processing, and correct preparation of food. It also has an educational component, as it aims to provide information, promote learning, and reinforce knowledge associated with eating habits and, complementarily, with healthy lifestyles. The diagnostic questionnaire is applied to all households that will be part of the program in order to obtain information from three main areas: family group socio-demographic characteristics, general food security determinants, and food diagnosis. We selected the items of each area that best matched this study’s aims, as shown in Table 1.

Until 2013, the FSPFS was focused on increasing family savings and security through self-production of food. Following an evaluation of the program, it was concluded that the saving capacity was difficult to measure and that it should shift focus to peoples’ “right to food”, paralleling a broader change of prioritization among those working for food security. Thus, from 2013 onwards, the program focused on safeguarding the consumption and diversity of food in the most vulnerable families in the country through the improvement of the availability of healthy foods and food education.

Each family responds to a specific diagnosis regarding their frequency of consumption for 12 food groups, which then lead to three types of recommendations: “below-recommended”, “recommended”, and “above-recommended”. Although the dietary recommendations for the Chilean population for the year 2009/2010 were taken as a reference, most are the result of the program’s own creation. For each of the 12 food groups, a frequency of consumption was defined in order to approximate, over a period of 1 month, dietary intake, and thus determine whether to work on a production technology if the consumption is below what is recommended, or focus on dietary education if consumption is above recommendations. The three categories mentioned above were created based on international standards and the previously mentioned dietary guidelines for the Chilean population and seek to make it easier for the executor, who is the figure that works directly with the families, to perform the job of uploading the information to the centralized system. However, the program recognizes that this type of question never provides information that is 100% accurate, since food guidelines should also be based on the territory, realities, and integrating cultural aspects. A total standardization can, among other things, end up suggesting a culturally or otherwise inappropriate way of asking about culinary or consumption patterns. Finally, one of the most recent modifications of the diagnostic instrument involved adding information on the consumption of the different food groups in terms of “quantity”, improving the accuracy provided by the information associated with the frequency of consumption.

In order to add the geographical characteristics to this study, we added to our dataset a column with the five Chilean macro-zones: Norte Grande, Norte Chico, Zona Central, Zona Sur, and Zona Austral, due to climate differences between each macro-zone. We wanted to see if there were territorial differences regarding the intake of the food groups amongst the macro-zones. Thus, each macro-zone groups the following regions, climate type according to the Köppen–Geiger classification, and agricultural production, as shown in Table 2.

### Methods Employed by Objective

Objective 1—Socio-Demographic Analysis of Vulnerable Households: The first step involved a comprehensive analysis of the FSPFS survey, focusing on households identified in 2019 and expanded to include additional data from 2020. The investigation centered around socio-demographic characteristics, such as household headship, gender of the family head, and number of underaged family members. The analysis was further deepened into food security determinants, including food availability issues, food access issues and water access, and territorial determinants, including all macro-zones.

Objective 2—Dietary Intake Frequency Assessment: For the evaluation of dietary habits, a quantitative approach was adopted, which was based on categorizing the intake frequency of n = 12 food groups, listed below, according to national dietary guidelines and intake recommendations.

VegetablesFruitsDairyWhite meatRed meatEggsLegumesWaterBreadCerealsFatSugar

The assessment stratified households based on “below recommended”, “recommended” and “above recommended” intake levels, further dissecting consumption patterns into weekly intake frequencies (never [0 times/week]; sometimes [0.5 times/week]; [1–2 times/week]; [3–5 times/week]; every day [7 times/week]) to provide a detailed portrayal of the dietary landscape among Chile’s vulnerable populations.

Objective 3—Dietary Diversity Score Calculation and Analysis: The HDDS served as a pivotal metric in this phase, calculated for a sizable cohort to explore correlations with socio-demographic variables, food security determinants, and regional disparities. This objective employed descriptive statistics for a national-level dietary diversity overview, while regional scores were scrutinized to pinpoint macro-zone dietary patterns. Multivariate regression techniques were applied to assess the impact of various socio-demographic and food security factors, alongside geographical considerations, on the HDDS.

The calculation of the HDDS for each household was conducted in several steps utilizing the data on weekly intake frequencies of n = 12 food groups from the FSPFS diagnosis. The methodology employed is outlined as follows:
a.Quantification of Intake Frequencies: The weekly intake frequency of each food group was recorded as HWFfgi (Household Weekly Frequency), where i represents each of the 12 food groups. The frequency was quantified on a scale based on the reported intake:
-Never [0 times/week] = 0-Twice per month [0.5 times/week] = 0.5-1–2 times/week = 1.5-3–5 times/week = 4-Every day [7 times/week] = 7

b.Weighting of Food Groups: Each HWFfgi was then multiplied by a weight factor Wi specific to each food group to reflect its importance in the diet, as shown in Table 3. The weight factors Wi were derived from [26] nutritional guidelines that emphasize the relative nutritional contribution of each food group:Weighted_HWFfgi=HWFfgi×Wi

c.Calculation of HDDS: The HDDS for each household was calculated by summing the weighted frequencies of all 12 food groups:
$$HDDS=\sum_{i=1}^{12}Weighted\_HWFfgi$$

The final HDDS is a summative score that indicates the dietary diversity of the household, serving as a proxy for nutritional adequacy. Higher HDDS values suggest greater dietary diversity and, potentially, better household nutritional status.

Using the “lm” function in R, a multilinear regression model with HDDS as the dependent variable was constructed. The model included Head of Household Age, Gender, Underaged family members, Food Access Issues, Food Availability Issues, Water Access, and Macro-zone as independent variables. To assess the collinearity among the variables in the dataset, a Variance Inflation Factor (VIF) test was employed by calculating VIF values for each predictor using the “vif” function from the “car” package. A VIF value exceeding 5 was considered indicative of significant collinearity.

Objective 4—Development of a Spatial Warning System: The data for this objective analysis were extracted from the diagnostic surveys from the FSPFS database. Based on Objective 2 of our preliminary analysis, which identified problematic food group intakes, five critical food groups were selected: fruits, vegetables, dairy products, sugars, and fats. These groups were chosen due to their significant deviation from the recommended intake levels—fruits, vegetables, and dairy intake were typically under the recommended thresholds, while sugars and fats were consumed above the recommended thresholds.

Descriptive statistics were computed for each of the five critical food groups across all macro-zones. For the data analysis, the RStudio environment was utilized with its packages “tidyverse”, “chilemapas”, “sf”, “ggplot2”, and “dplyr”. The monthly intake averages were calculated to determine the mean number of days each food group was consumed. To capture the variability in consumption patterns, we identified the highest and lowest intake frequencies, along with the corresponding number and percentage of households. The mode representing the most common reported intake frequency was also calculated along with its prevalence among the households. To quantify the degree of intake deficiency, we determined the percentage of households consuming at/below/above the recommended levels for each food group.

The spatial visualization analysis of the five critical food groups’ intakes across macro-zones in Chile involved the following steps of data processing and geographical mapping:
a.Conversion of Weekly Intake Frequency to Monthly Frequency:

Since the data provided by the FSPFS diagnosis survey considered a weekly intake, the function MonthlyIntake(i) was defined to convert weekly frequency intake to monthly frequency intake.
$$\left\{\begin{array}{c} 30\;if\;Everyday\\ 16\;if\;3-5\;times\;per\;week\\ 6\;if\;1-2\;times\;per\;week\\ 2\;if\;Sometimes\\ 0\;if\;Never \end{array}\right.$$
b.Calculation of Commune-Level Averages

For each commune, i.e., the smallest administrative division in Chile, the average monthly frequency intake, defined as CommuneAverage(i) was calculated, by aggregating individual household monthly frequency intakes for a given food group i and then dividing by the total number of households in the commune.
$$CommuneAverage\left(i\right)=\;\frac{\sum MonthlyIntake(i)}{Total\;number\;of\;households\;in\;the\;commune}$$
where n is the number of households in the commune.

Each commune’s *CommuneAverage*(*i*) was georeferenced using its coordinates provided by the “chilemapas” package in RStudio. This process involved associating the calculated average intake values with their respective spatial locations on the Chilean map.

A Choropleth map was applied to represent the average intake values. Communes with higher average intakes of a particular food group are displayed in darker shades, while those with lower average intakes are shown in lighter shades. This gradient visually represents the distribution of intake frequencies across different communes and macro-zones.

We identified and grouped communes within the macro-zones based on the proportion of households with intake levels below the recommended threshold for fruits, vegetables, and dairy, and above the recommended threshold for fats and sugars. The benchmark for categorization was set at 50% or more of households deviating from the recommended intake levels.

For visual representation, we employed a horizontal barplot with a red gradient fill, where the intensity of the red color corresponded to the proportion of households in each commune with above or below the recommended dietary intake. The gradient provided a visual scale of adherence to dietary recommendations, with darker shades indicating a higher percentage of households not meeting the recommended intake. The analysis was designed to yield both visual and quantitative interpretations of dietary patterns, facilitating the development of targeted nutritional interventions. The formula employed for determining the thresholds was
(1)$$Threshold\;Proportion=\;\frac{Number\;of\;households\;not\;meeting\;recommendations}{Total\;number\;of\;households\;in\;the\;commune}\times 100$$
where a proportion equal to or greater than 50% indicated a significant deviation from recommended dietary patterns. The proportion of households not meeting the recommendations per food group in different macro-zones can be found in Appendix A.

## 3. Results

### 3.1. Socio-Demographic Analysis of Vulnerable Households

#### 3.1.1. Household Composition, Size, and Type

It was found that all necessary data were available for n = 3967 households, from a total of n = 3996 households initially included for 2019, meaning an inclusion rate of 99.3%. In addition, six communes from 2020 were incorporated in the analysis, resulting in a total of n = 4047 households and n = 12,534 individuals. Of these, n = 3505 (86.6%) households had a woman as head, whereas n = 542 households (13.4%) were led by men. On average, each household had 3.05 members: 3.2 members when the household was led by a woman, and 2.15 members when it was led by a man. When analyzing this variable in the different macro-zones, while in all cases there were more female heads of households, in the Norte Grande, the difference was 77.0% vs. 23.0%, whereas the largest difference was found in the Zona Central, with 89.2% vs. 10.8%. At the national level, it was found that small households were the most frequent, comprising 63.8% of the total, followed by medium-sized households (32.9%), and large households (3.3%). Small households consist of up to 3 members, medium-sized households consist of 4–6 members, and large households consist of 7 or more members. This trend was consistent across all macro-zones, although with variations. For example, in the most extreme zones (i.e., northern and southern regions), Norte Grande and Zona Austral, smaller households accounted for 86.6% and 78.6% of the total, while larger households constituted only 1.7% and 2%, respectively. Refer to Table 4 for detailed statistics on household composition, headship by gender, and size across different macro-zones in Chile.

#### 3.1.2. Age and Regional Variations

The national average age for men was 68.9 ± 12.1 years, and for women, it was 55.9 ± 15.6 years, representing a difference of 13 years. The highest average ages were found in the Zona Austral, while the lowest were in the Norte Chico. It is notable that the groups aged 18 to 64, often considered the workforce group, were the most frequent, followed by individuals older than 64 years. Regional variations in age distributions should be noted as well. Refer to Table 5 For detailed information on the average age of household heads by gender and the distribution of age groups across different Chilean macro-zones.

#### 3.1.3. Employment Status and Main Work Activities

Regarding the employment status of household heads, there was a noticeable difference between households led by women and men. It was found that 25.4% (796 individuals) of women who were household heads were employed, while 37.45% (298 individuals) of men who were household heads were employed. When analyzing the main work activities of the family representative, it was found that, for women, the main activities were homemaker (59.86%), pensioner (12.21%), and farmer/food production (9.64%). In the case of households headed by men, the primary activities were pensioner (42.25%), farmer/food production (29.52%), and other occupations (17.34%). For the purposes of this study, homemaker and unemployed were considered mutually exclusive categories. Refer to Table 6 for more details on occupation according to head of household.

### 3.2. Dietary Intake Frequency Assessment

#### Intake of Different Food Groups among Chilean Vulnerable Households at the National Level

When analyzing the intake of different food groups among Chilean vulnerable households according to the “recommendation” categories, it was possible to highlight different tendencies (Appendix A) all shown schematized in Figure 1. First, it was found that recommended intake of food groups such as bread, cereals, eggs, legumes, sugar, water, and white meats were each met or exceeded by at least 75% of the households. Secondly, not all food groups had intakes in the “above recommended” category; these food groups included vegetables, fruits, dairy and water. In addition, only in the case of vegetables, fruits and dairy, there were more households with “below recommended” intake in comparison to those with “recommended” and “above recommended” intake. Finally, the only food groups that were regularly overconsumed were fats and sugars.

However, the above describe the proportion of households within each recommendation category and do not give any detail regarding the frequency of intake for the different food groups. Thus, Figure 2 shows this information according to the intake frequency detailed in Appendix A. One of the main findings in this analysis is related to the intake of bread, which is consumed daily by 92% of households. It was also found that nearly half of the households consumed vegetables and cereals on a daily basis. Also, fats and sugar were consumed at least once a week by all households. Most households consumed protein at least once per week, mainly through white meat and/or legumes. Finally, most members of households consumed water daily.

### 3.3. HDDS Calculation and Analysis

#### 3.3.1. Chilean Vulnerable HDDS

##### National Level

The 4047 observations of the HDDS showed a mean value of 91.4 ± 20.63, indicating a noticeable variability in the dietary diversity across households. The median value of 91.75 closely aligns with the mean, suggesting a symmetric distribution. The range from a minimum score of 20 to a maximum of 163.25 showcases considerable variability in dietary diversity scores. The slight negative skewness of −0.064 indicates that the distribution is slightly skewed to the left, although this skewness is minimal. The HDDS histogram in available in Figure 3.

##### Macro-Zone Level

The HDDS data provide a comprehensive overview of dietary diversity among Chile’s macro-zones, as schematized in Figure 4. The mean HDDS indicates moderately diverse diets in Norte Grande (96.07 ± 19.23) and Zona Central (94.15 ± 19.81), while Norte Chico (93.35 ± 19.03) and Zona Austral (91.26 ± 20.56) also exhibit reasonable diversity. Zona Sur shows a somewhat lower mean HDDS at 88.1 ± 21.28, suggesting a somewhat less diverse diet. The standard deviations reflect the variability within these regions. Median values closely align with the means, indicating relatively balanced distributions. Zona Central stands out with a mode of 110, suggesting a substantially greater HDDS. The interquartile range (IQR) demonstrates the spread of data, with Zona Central having the widest range (20.00 to 163.25), while Zona Sur exhibits the narrowest range (30.50 to 158.25). Details for all macro-zones can be found in Table 7.

#### 3.3.2. HDDS Association with Household Socio-Demographic Characteristics, Food Security Determinants, and Geographical Determinants

The multiple linear regression analysis conducted to evaluate the relationship between the HDDS and socio-demographic characteristics, food security and geographical determinants showed significance in the overall model (R^2^ adjusted = 0.0284, F_10,4036_ = 12.84, *p* < 0.001). The HDDS was positively correlated with number of minors in the household (*β* = 0.969, *p* = 0.005) and water access (*β* = 1.410, *p* = 0.050). In contrast, having food access issues (*β* = −2.267, *p* = 0.002) and residing in the ZS macro-zone, compared to NC as reference (*β* = −4.976, *p* < 0.001), were negatively associated with the HDDS. Other variables, namely age, gender, and food availability issues, did not show significant associations (Table 8).

The calculated GVIF values, as shown in Table 9, confirm that no variable exhibits significant collinearity, supporting the robustness of the regression model.

### 3.4. Development of a Spatial Warning System

#### 3.4.1. Food Group Intake in Vulnerable Households According to Geographical Macro-Zones Including “Norte Grande”, “Norte Chico”, “Zona Central”, “Zona Sur”, and “Zona Austral”

##### Fruits

The highest averages for fruit consumption were found in the Zona Central (19.8 ± 10.1) and Norte Chico (18.0 ± 9.7), whereas the lowest were found in the Zona Sur (16.3 ± 10.2) and the Zona Austral (15.9 ± 9.4). Although there was a difference of almost 4 days per month between the macro-zones with the highest and lowest intake, it was found that all macro-zones had, on average, an intake of once every two days. All macro-zones showed a daily intake of fruits in at least one of their communes. However, n = 8 communes in the Zona Central (0.48%), n = 4 (0.2%) communes in the Zona Sur, and n = 5 (5.1%) communes in the Zona Austral showed no intake of fruit during the month. Finally, whereas a daily intake of fruits was found as the more frequent value in the Norte Grande and Zona Central, in the Zona Austral, the most repeated value, representing 28% of the households, was an intake of 6.4 days/month. Regarding the number of households below the recommended fruit intake, the southern macro-zones showed the highest values, with the Zona Austral (n = 81) reaching 82.6% of households under that condition, and Zona Sur (n = 1290) 78.1% of households under the same classification. Details about the values of these indicators for each region and the spatial representation of these differences can be found in Table 10 and Figure 5, respectively.

##### Vegetables

The Zona Central (23.81 ± 8.8) and Norte Grande (21.2 ± 9.5) were the macro-zones with the highest monthly average intake of vegetables per household, whereas the Zona Austral (18.1 ± 8.9) and the Norte Chico (19.9 ± 9.8) were the macro-zones with the lowest monthly average intake. All macro-zones had at least one household with an average intake of vegetables of 30 days/month, with the Zona Central (n = 1058; 63.4%) being the macro-zone with the highest number, and the Zona Austral (n = 29; 29.6%) the macro-zone with the lowest. However, it was only in the Zona Sur where we found households (n = 2; 0.1%) with no consumption of vegetables. Finally, except for the Zona Austral, whose most frequent vegetable intake average was 17.1 days/month in n = 38 (39%) of their households, all other macro-zones had a daily intake of vegetables as their most frequent value, compared to the other intake frequencies. As with fruit intake, the southern macro-zones were those with the greatest proportion of households in the “below recommended” category (Zona Sur: n = 1128 (63.5%); Zona Austral n = 69 (70.4%)). Details about the values of these indicators for each region and the spatial representation of these differences can be found in Table 11 and Figure 6, respectively.

##### Dairy

The highest averages for dairy intake per household were found in the Norte Grande (19.2 ± 10.8) and the Zona Central (18.5 ± 10.9), whereas the lowest averages were in the Zona Austral (15.9 ± 11) and the Zona Sur. All macro-zones had at least one household that showed a daily intake of dairy products, with the Norte Grande (44%) and the Zona Central (41.8%) being those with a major proportion. However, all macro-zones had a daily intake of dairy products as their most frequent value. In addition, at least one household in all macro-zones showed a zero intake of dairy products within the monthly period, most commonly in the Zona Central (n = 51; 3.05%) and the Zona Austral (n = 5; 5.1%). For this food group, 72.3% (n = 1284), of Zona Sur households were below the recommended intake of dairy products, whereas 69.4% (n = 68) of households in the Zona Austral fell into this classification. Details about the values of these indicators for each region and the spatial representation of these differences can be found in Table 12 and Figure 7, respectively.

#### Fat

The Norte Grande (9.8 ± 10.7) and the Norte Chico (10.5 ± 10.2) were the macro-zones with the highest average intake of fat. All macro-zones had at least one household with a daily intake of fat: the Norte Grande (n = 31; 17.4%) and the Norte Chico (n = 46; 15.4%) had the highest values. Similarly, all macro-zones showed at least one household with no intake of fat at all. Of these, the Norte Grande (n = 21; 11.8%) and the Zona Austral (n = 10; 10.2%) were the macro-zones with major representation. Finally, the most frequent value for household fat intake across all macro-zones was 2.1 days/month. The northern macro-zone had the highest values regarding the overconsumption of fat. Thus, the Norte Chico presented 56% (n = 167) of households under this classification, whereas the Norte Grande had a rate of 50.6% (n = 90). Details about the values of these indicators for each region and the spatial representation of these differences can be found in Table 13 and Figure 8, respectively.

##### Sugar

The macro-zones with the highest average sugar intake were the Norte Grande (10.5 ± 11.1) and the Zona Central (10.5 ± 11.3). All macro-zones had at least one household with a daily intake of sugar, with the Zona Central (n = 363; 21.8%) and the Norte Grande (n = 36; 20.2%) being those with the highest values. Similarly, all macro-zones included some households with no intake of sugar. In this case, the Zona Austral (n = 11; 11.2%) and the Zona Sur (n = 192; 10.1) had the highest values. For all macro-zones, the most frequent household sugar intake was 2.1 days/month. The Norte Grande (n = 88; 49.4%), Norte Chico (n = 149; 50%), and Zona Central (n = 840; 49.5%) presented the highest rates for households with overconsumption of sugar. Details about the values of these indicators for each region and the spatial representation of these differences can be found in Table 14 and Figure 9, respectively.

## 4. Discussion

The results of this research indicate significant differences in the HDDS based on the number of underage family members, access to water, food access issues and residing in the Zona Sur macro-zone. Additionally, although the multivariate regression analysis showed that the gender of the head of household was not significantly related to the HDDS, the reality is that 86.6% of the households in the analyzed sample were led by women. Therefore, there is a need for special attention to how gender-focused food policies are addressed These relationships between socio-demographic factors and dietary diversity in Chile reflect global trends. This is further contextualized by research from Honduras [2], suggesting the unique challenges of female-headed households, a demographic that is prevalent in Chile and that requires targeted interventions. In this sense, these findings could mean a first step to deepen analysis into women’s diet quality and micronutrient status in developing countries to better inform interventions and policies, as suggested by [4]. Similarly, in South Africa [27], the link between education level and dietary diversity, as the approach followed by the FSPFS, emphasizes the potential for educational initiatives to enhance food security in Chile. These global insights highlight the importance of contextualized, gender-sensitive, and education-focused strategies to address the dietary diversity and food security nexus within Chilean households.

While the HDDS is a valuable tool for identifying food security and nutritional needs, it is important to note that the model explained approximately 2.84% of the variability in the HDDS. This indicates that other factors not included in the analysis might be influencing household dietary diversity. For example, Ref. [5] noted that the HDDS fails to capture actual food consumption amounts, could be subject to reporting bias, and might not be representative due to its cross-sectional nature. Ref. [6] also pointed to the potential for measurement error given its reliance on self-reported data and its lack of granularity regarding the quantity of food within groups. Ref. [2] echoed this, emphasizing the score’s inability to account for the specific nutritional content of foods. Ref. [7] added concerns about potential biases and the lack of empirical evidence linking dietary diversity with energy intake. Lastly, Ref. [27] criticized the HDDS for not considering intra-household food distribution inequalities and the difficulty in comparing HDDSs across different studies due to variations in food groupings. Furthermore, the HDDS is a number that can be reached through different combinations of food group consumption, so it must also be viewed with caution.

The prevalent consumption patterns, particularly the under-consumption of fruits, vegetables, and dairy, and the over-consumption of fats and sugars, have direct implications for the nutritional health of vulnerable populations. The dietary patterns observed in Chile, characterized by the under-consumption and over-consumption of certain food groups, mirror global concerns highlighted in international research. In South Africa, Ref. [27] noted that despite high dietary diversity, there is a lack of micronutrient-rich foods, a pattern that might be reflected in Chile’s vulnerable populations, who also demonstrate gaps in essential nutrient intake. This is compounded by the findings from Honduras, which indicate that despite adequate caloric intake, nutrient-rich dietary diversity is lacking [2], a concern that is likely paralleled in Chile. Similarly, Ref. [7] in Myanmar reported low dietary diversity with negative health implications, echoing the potential nutrient deficiencies among the Chilean populace. In Malawi, Ref. [6] suggested that increased dietary diversity correlates with better food security, a strategy that could be beneficial for Chile. Lastly, the socioeconomic factors affecting dietary habits in Bangladesh, as noted by [5], resonate with the Chilean context, where economic stresses, particularly due to the COVID-19 pandemic, may exacerbate dietary limitations, affecting health outcomes.

The results also show a centralization of healthy eating in Chile, highlighting regional disparities in the consumption of healthy foods in the vulnerable population. In the Zona Central, there is a higher frequency of fruit and vegetable consumption, with a monthly average of 19.8 ± 10.1 days and 23.8 ± 8.8 days, respectively. In contrast, the extreme regions, such as the Norte Grande and the Zona Austral, face significantly different realities. The Norte Grande, with a monthly average consumption of fats and sugar of 9.8 ± 10.7 days and 10.5 ± 11.1 days, respectively, shows an unhealthy consumption profile. The Zona Austral, on the other hand, has the lowest consumption of fruits (15.9 ± 9.4 days), vegetables (18.1 ± 8.9 days), and dairy (15.9 ± 11 days). These disparities are partly due to the geographical and climatic conditions of the different macro-zones [24]. The Zona Central, being the main area for fruit and vegetable production [25], facilitates access to fresh produce. In contrast, the Norte Grande, an arid region, and the Zona Austral, with conditions that do not favor mass production of fruits and vegetables [24,25], face significant challenges in accessing a balanced diet. This situation underscores the need to improve food governance and ensure equitable physical access to healthy foods across the country.

In addition, there is a potential of the geospatial warning system proposed in pinpointing areas of food insecurity, representing a good tool for the FSPFS to know where the most critical areas are to better localize the resources needed. From this perspective, the literature widely recognizes the potential of these systems. For example, Ref. [13] validated its precision in identifying vulnerable districts, aiding in resource allocation and planning towards Sustainable Development Goals. Ref. [16] emphasized its role in revealing spatial distribution and trends in Rajasthan, directing targeted interventions. Ref. [14] highlighted mapping’s capacity to elucidate spatial patterns, although they caution that its success hinges on data quality and stakeholders’ interpretative skills. Similarly, Ref. [15] demonstrated how mapping informs unmet needs, optimizing efforts to combat food insecurity in New Hampshire. However, some important limitations in this study are necessary to acknowledge. First, the use of a single-time data collection point during the diagnostic survey phase presents a “picture”, limiting the ability to observe changes over time or to continuously integrate new data. This is compounded by the reliance on the HDDS, which, although useful, cannot encapsulate the nuances of nutritional status or food consumption patterns over time. Additionally, the program’s focus on new families each year complicates the ability to track longitudinal progress, with the only measure of continuity being the communal, regional, or macro-zone data trends.

While the spatial analysis focuses on individual food groups, which provide valuable information, it would be beneficial to understand how these food groups interact within vulnerable contexts. Specifically, analyzing “meals” rather than just ingredients could offer a more comprehensive view of dietary practices. This approach would allow for a better understanding of the nutritional balance and cultural relevance of food consumption patterns in these populations. Considering meals as holistic units rather than isolated components could reveal insights into dietary habits, food security, and nutritional outcomes that are otherwise obscured by looking at ingredients alone. By shifting the focus to meals, interventions could be more effectively tailored to meet the actual dietary needs and preferences of vulnerable groups.

## 5. Conclusions

This study highlights that food insecurity remains a critical issue in Chile, particularly affecting vulnerable households as these exhibit limited dietary diversity, with insufficient intake of essential food groups like fruits, vegetables, and dairy, while consuming excessive fats and sugars. The analysis of intake frequency and the HDDS revealed that sociodemographic factors, food access issues, and water availability significantly influence dietary patterns. A notable sociodemographic finding is the gender disparity, with 86.6% of households led by women, indicating a need for gender-sensitive policies related to food. The study also observed that average age differences between male and female household heads suggest potential age-related vulnerabilities. Additionally, small households in extreme regions highlight specific resource needs, while employment patterns reveal that housework, pensions, and family farming are the predominant work activities, providing insight into livelihood sources.

The practical recommendations based on the results and analysis of this study are as follows: (1) When selecting households that could participate in the program, it is essential to analyze their demographic characteristics to ensure that the inclusion of these households is based on identifying social conditions that make them more vulnerable to having lower food diversity. (2) Understanding the distribution of consumption of critical food groups across Chilean territory can also aid in the prior planning of the logistics required to improve access to these foods in the participating households.

While this study provides a comprehensive view of dietary diversity in vulnerable Chilean households, it is important to acknowledge its limitations. The single-time data collection limits the ability to observe changes over time. Furthermore, reliance on the HDDS, though useful, does not fully capture the complexities of nutritional status and food consumption patterns. The prevalence of concerning consumption patterns, such as the under-consumption of fruits, vegetables, and dairy, and the over-consumption of fats and sugars, has direct implications for the nutritional health of vulnerable populations.

Recommendations for future studies include longitudinal tracking of households to assess changes in dietary diversity and nutritional status, as well as analyzing the interactions between different food groups within vulnerable contexts. Additionally, further investigation into how specific sociodemographic characteristics, such as the gender of the household head and family composition, influence food security and dietary diversity is suggested. Understanding these regional variations is vital for tailoring targeted interventions to address the specific dietary needs of each macro-zone.

## Figures and Tables

**Figure 1 nutrients-16-02937-f001:**
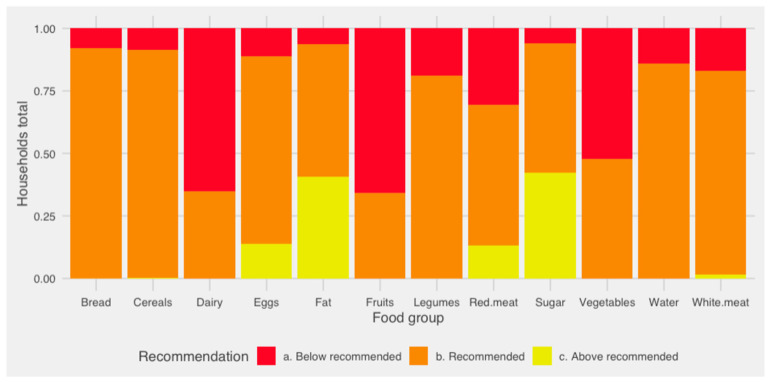
Recommended intake of different food groups in vulnerable Chilean households, 2019–2020.

**Figure 2 nutrients-16-02937-f002:**
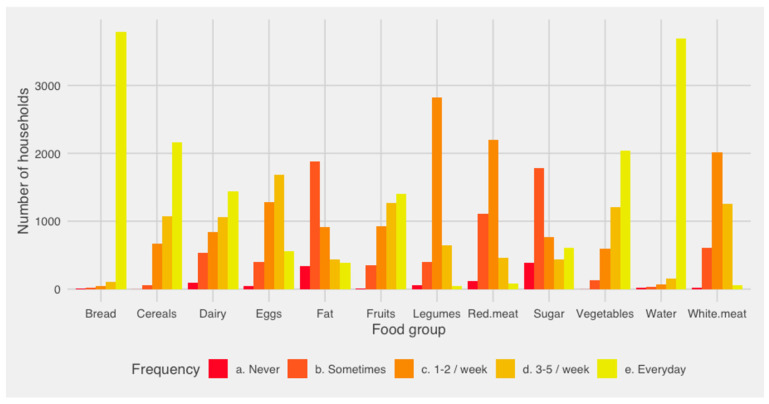
Frequency of intake of different food groups in vulnerable Chilean households, 2019–2020.

**Figure 3 nutrients-16-02937-f003:**
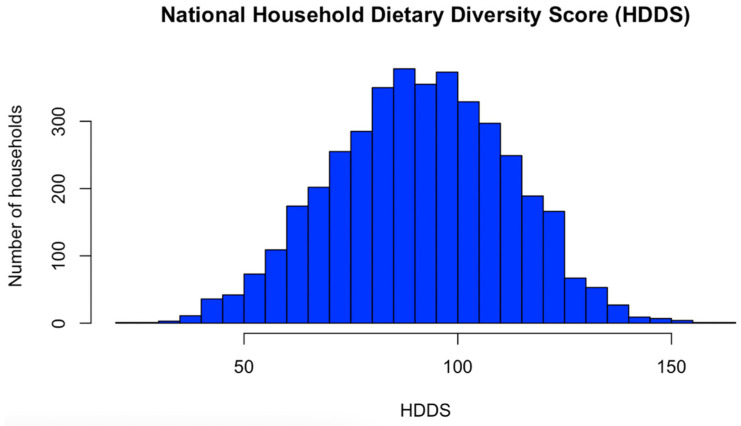
National Household Dietary Diversity Score (HDDS), 2019–2020.

**Figure 4 nutrients-16-02937-f004:**
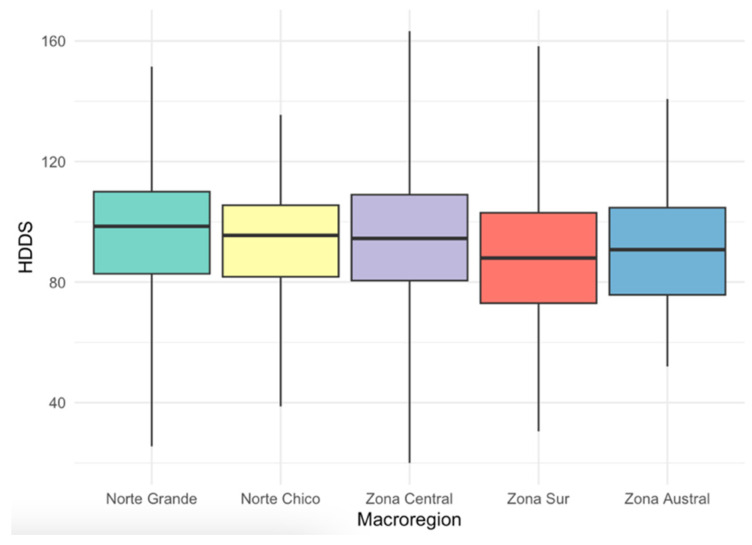
Distribution of HDDS across Chilean macro-zones, 2019–2020.

**Figure 5 nutrients-16-02937-f005:**
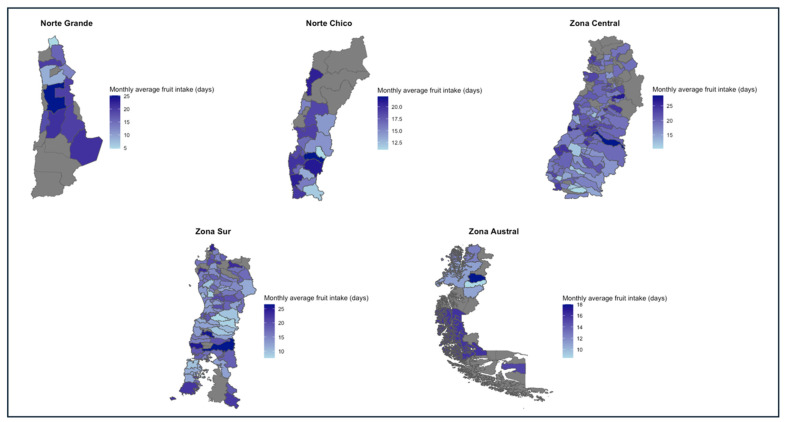
Monthly average fruit intake in days per macro-zone, 2019–2020.

**Figure 6 nutrients-16-02937-f006:**
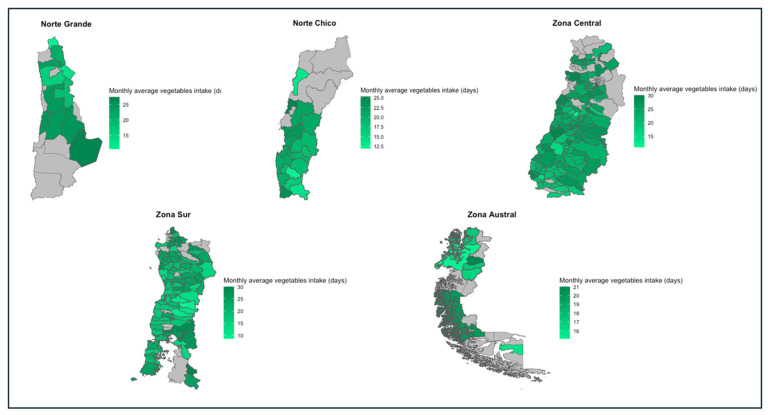
Monthly average vegetable intake in days per macro-zone, 2019–2020.

**Figure 7 nutrients-16-02937-f007:**
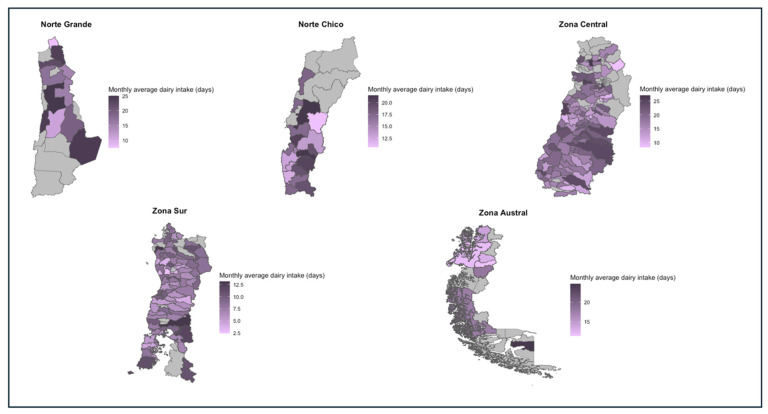
Monthly average dairy intake in days per macro-zone, 2019–2020.

**Figure 8 nutrients-16-02937-f008:**
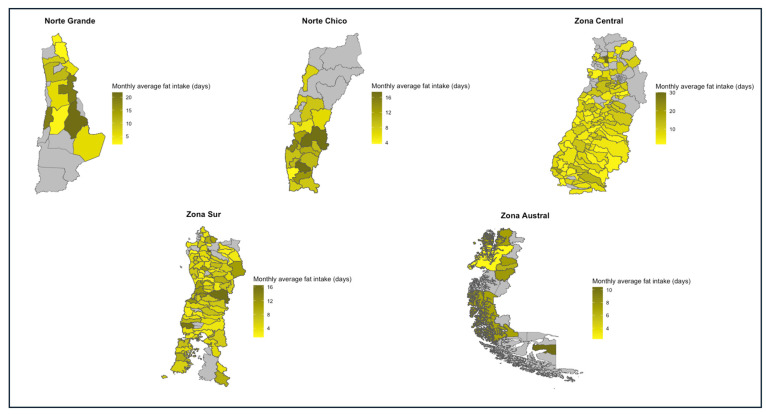
Monthly average fat intake in days per macro-zone, 2019–2020.

**Figure 9 nutrients-16-02937-f009:**
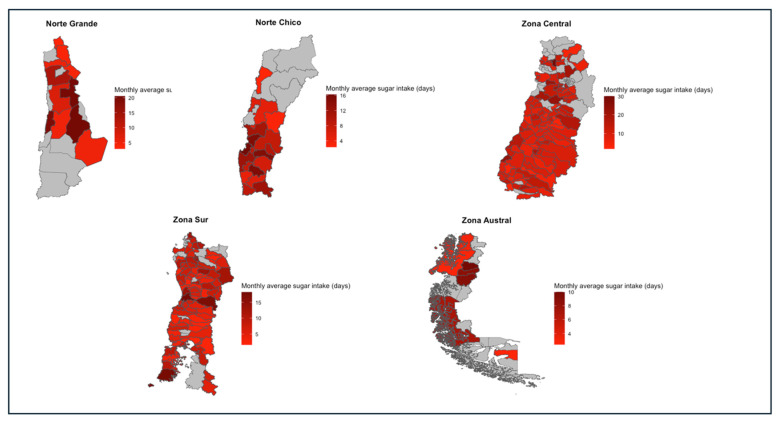
Monthly average sugar intake in days per macro-zone, 2019–2020.

**Table 1 nutrients-16-02937-t001:** Diagnostic questionnaire content.

Area	No of Items	Items
Family group socio-demographic characteristics	11	Region; commune; head of household (Hoh) gender; (Hoh) age; (Hoh) main work activity; 0–5 years old; 6–9 years old; 10–17 years old; 18–64 years old; 65+ years old; total family members.
General food security determinants	4	Food access issues; food availability issues; water access issues; pollution free environment.
Food diagnosis	24	“Weekly intake frequency” and “below recommended/recommended/above recommended” for the following food groups: vegetables, fruits, dairy, white meat, red meat, eggs, legumes, water, bread, cereals, fat, sugar.

**Table 2 nutrients-16-02937-t002:** Chilean regions grouped in macro-zones (adapted from Sarricolea et al., 2017 [24] and Meza et al., 2021 [25]).

Macro-Zone	Regions	Climate Type	Agricultural Production
Norte Grande	Arica y Parinacota, Tarapacá, Antofagasta	Arid and polar climates due to the Atacama Desert and high altitude.	Smallholder farming: horticultural crops (corn, lettuce, tomato, bell pepper). Fruit trees: citrus, mangoes, olives
Norte Chico	Atacama, Coquimbo	Arid but also experiences polar climates at higher elevations.	Table grapes, wine grapes, mandarins, avocados, blueberries.
Zona Central	Valparaíso, Metropolitana, O’Higgins, Maule, Ñuble, Biobío	The primary agricultural zone, where temperate climates cover over 90% of the region.	Export-oriented agriculture: fruits (grapes, apples, berries), vegetables. Large-scale farming systems with diverse crops.
Zona Sur	La Araucanía, Los Lagos, Los Ríos	Temperate climate with minor tundra zones.	Transition to export-oriented fruit trees: walnuts, blueberries, hazelnuts, cherries. Traditional crops: pastures, wheat, barley. Dairy and beef farming.
Zona Austral	Aysén, Magallanes	Mix of polar and temperate climates, reflecting the cold and humid conditions.	Limited agricultural activity. Focus on livestock, mainly sheep, and forestry.

**Table 3 nutrients-16-02937-t003:** Weight factors per food group.

Food Group	Weight Factor
Vegetables	1.0
Fruits	1.0
Dairy	4.0
White meat	4.0
Red meat	4.0
Eggs	4.0
Legumes	3.0
Water	1.0
Bread	2.0
Cereals	2.0
Fat	0.5
Sugar	0.5

**Table 4 nutrients-16-02937-t004:** Household composition, headship by gender and size across Chilean macro-zones, 2019–2020 *.

Macro-Zone	Households Analyzed	Households with Female Head	Households with Male Head	Average Household Size	Small Households	Medium-Size Households	Large Households
Norte Grande	178 (4.4%)	137 (77.03%)	41 (23.03%)	3.1	111 (62.36%)	56 (31.46%)	11 (6.18%)
Norte Chico	298 (7.4%)	265 (88.93%)	33 (11.07%)	3.1	180 (60.4%)	105 (35.23%)	13 (4.36%)
Zona Central	1668 (41.2%)	1488 (89.21%)	180 (10.79%)	3.1	1052 (63.07%)	565 (33.87%)	51 (3.06%)
Zona Sur	1805 (44.6%)	1533 (84.93%)	272 (15.07%)	3.1	1163 (64.43%)	585 (32.41%)	57 (3.16%)
Zona Austral	98 (2.3%)	82 (83.67%)	16 (16.33%)	2.5	77 (78.57%)	19 (19.39%)	2 (2.04%)

* Note: Percentages may not sum to 100 due to rounding.

**Table 5 nutrients-16-02937-t005:** Average age of household heads and age group distribution by gender in Chilean macro-zones, 2019–2020 *.

Macro-Zone	Average Age of Male Heads (Years)	Average Age of Female Heads (Years)	Percentage of Workforce Age Group (%)	Percentage of Elderly (65+)
Norte Grande	67.9	51.2	39.6	18.6
Norte Chico	66.5	54.1	50.6	13.7
Zona Central	69.3	55.2	51.1	16.4
Zona Sur	68.5	57	49.5	20.8
Zona Austral	76.5	61.1	42.9	29

* Note: The workforce age group is defined as individuals aged 18 to 64 years.

**Table 6 nutrients-16-02937-t006:** Occupation according to head of household, 2019–2020.

Head of Household Gender	Occupation					
	Farmer/Food Production	Homemaker	Pensioner	Unemployed	Other	No Information
Female	338 (9.64%)	2098 (59.86%)	428 (12.21%)	54 (1.54%)	558 (15.92%)	29 (0.83%)
Male	160 (29.52%)	43 (7.93%)	229 (42.25%)	5 (0.92%)	94 (17.34%)	11 (2.03%)

**Table 7 nutrients-16-02937-t007:** Household Dietary Diversity Score (HDDS) across Chilean macro-zones, 2019–2020.

Macro-Zone	Mean ± SD	Median	Mode	Q25	Q75	Min	Max
Norte Grande	96.07 ± 19.23	98.50	90.0	82.75	110	25.50	151.50
Norte Chico	93.35 ± 19.03	95.50	97.0	81.75	105.5000	38.75	135.50
Zona Central	94.15 ± 19.81	94.50	110.0	80.50	109.0000	20.00	163.25
Zona Sur	88.1 ± 21.28	88.00	95.0	73.00	103	30.50	158.25
Zona Austral	91.26 ± 20.56	90.75	76.5	75.75	104.6875	52.00	140.75

**Table 8 nutrients-16-02937-t008:** HDDS association with household socio-demographic characteristics, food security determinants, and geographical determinants, 2019–2020.

		HDDS				
Variables		*n*	%	*β*	*t*-Value	*p*-Value
Socio-demographic	Head of household age	-	-	0.011	41.582	0.672
	Gender of family head					
	M	542	13.4%	0.896	0.423	0.362
	F	3505	86.6%	-	-	-
	Underaged family members	-	-	0.969	2.791	0.005 *
Food security determinants	Access issues					
	Yes	1943	48.0%	−2.26	−3.080	0.002 *
	No	2104	52.0%	-	-	-
	Availability issues					
	Yes	1805	44.6%	−1.28	−1.714	0.087
	No	2242	55.4%	-	-	-
	Water access					
	Yes	2786	68.8%	1.4	−1.962	0.05 *
	No	1261	31.2%	-	-	-
Geographical determinants	Macro-zone					
	Norte Grande	178	4.4%	3.082	1.592	0.111
	Norte Chico	298	7.4%	-	-	-
	Zona Central	1668	41.2%	0.872	0.682	0.495
	Zona Sur	1805	44.6%	−4.98	−3.874	0.0001 *
	Zona Austral	98	2.4%	−1.90	−0.798	0.425

Note: *n*: number of households; %: percentage of households of the total; *β*: regression coefficient. * *p* < 0.05.

**Table 9 nutrients-16-02937-t009:** Variance inflation factor (VIF) analysis for predictor variables, 2019–2020.

Variable	VIF
Head of household age	1.804657
Head of household gender	1.098242
Underaged family members	1.750329
Food access issues	1.323603
Food availability issues	1.352717
Water access	1.083933
Macro-zone	1.111615

**Table 10 nutrients-16-02937-t010:** Fruit intake across Chilean geographical macro-zones, 2019–2020.

Fruit	Norte Grande	Norte Chico	Zona Central	Zona Sur	Zona Austral
Monthly intake average (days)	17.5 ± 10.7	18 ± 9.7	19.8 ± 10.2	16.3 ± 10.2	15.9 ± 11
Highest (days)	30	30	30	30	30
n (%) [Households]	64 (36)	98 (32.9)	746 (44.7)	562 (31.1)	31 (31.6)
Lowest (days)	2.1	2.1	0	0	0
n (%) [Households]	25 (14)	16 (5.4)	8 (0.48)	4 (0.2)	5 (5.1)
Mode (days)	30	17.1	30	17.1	6.4
n (%) [Households]	64 (36)	110 (37)	739 (44)	596 (33)	27 (28)
No. of households below recommended * (%)	113 (63.4)	199 (66.8)	938 (55.2)	1290 (78.1)	81 (82.6)

* Details of communes with more than 50% of households below recommended intake can be found in Appendix A.

**Table 11 nutrients-16-02937-t011:** Vegetable intake across Chilean geographical macro-zones, 2019–2020.

Vegetables	Norte Grande	Norte Chico	Zona Central	Zona Sur	Zona Austral
Monthly intake average (days)	21.2 ± 9.5	19.8 ± 9.8	23.8 ± 8.8	20.2 ± 9.6	18.1 ± 8.9
Highest (days)	30	30	30	30	30
n (%) [Households]	86 (48.3)	128 (43)	1058 (63.4)	784 (43.4)	29 (29.6)
Lowest (days)	2.1	2.1	2.1	0	6.4
n (%) [Households]	9 (5)	12 (4)	29 (1.7)	2 (0.1)	26 (6.4)
Mode (days)	30	30	30	30	17.1
n (%) [Households]	86 (48)	127 (43)	1053 (63)	767 (42)	38 (39)
No. of households below recommended (%) *	93 (52.2)	175 (58.7)	646 (38)	1128 (63.5)	69 (70.4)

* Details of communes with more than 50% of households below recommended intake can be found in Appendix A.

**Table 12 nutrients-16-02937-t012:** Dairy intake across Chilean geographical macro-zones, 2019–2020.

Dairy	Norte Grande	Norte Chico	Zona Central	Zona Sur	Zona Austral
Monthly intake average (days)	19.2 ± 10.8	17.4 ± 10.4	18.5 ± 10.9	15.6 ± 11	15.9 ± 11
Highest (days)	30	30	30	30	30
n (%) [Households]	79 (44.4)	101 (33.9)	697 (41.8)	524 (29)	31 (31.6)
Lowest (days)	0	0	0	0	0
n (%) [Households]	3 (1.7)	2 (0.7)	51 (3.05)	40 (2.2)	5 (5.1)
Mode (days)	30	30	30	30	30
n (%) [Households]	79 (44)	103 (35)	694 (42)	542 (30)	39 (40)
No. of households below recommended (%) *	101 (56.7)	198 (66.4)	988 (58.1)	1284 (72.3)	68 (69.4)

* Details of communes with more than 50% of households below recommended intake can be found in Appendix A.

**Table 13 nutrients-16-02937-t013:** Fat intake across Chilean geographical macro-zones, 2019–2020.

Fat	Norte Grande	Norte Chico	Zona Central	Zona Sur	Zona Austral
Monthly intake average (days)	9.8 ± 10.7	10.5 ± 10.2	7.9 ± 9.4	6.1 ± 7.6	6.7 ± 8
Highest (days)	30	30	30	30	30
n (%) [Households]	31 (17.4)	46 (15.4)	197 (11.8)	118 (6.5)	7 (7.1)
Lowest (days)	0	0	0	0	0
n (%) [Households]	21 (11.8)	15 (5)	166 (10)	137 (7.6)	10 (10.2)
Mode (days)	2.1	2.1	2.1	2.1	2.1
n (%) [Households]	63 (35)	109 (37)	767 (46)	940 (52)	48 (49)
No. of households above recommended (%) *	90 (50.6)	167 (56)	741 (43.6)	603 (34)	46 (46.9)

* Details of communes with more than 50% of households above recommended intake can be found in Appendix A.

**Table 14 nutrients-16-02937-t014:** Sugar intake across Chilean geographical macro-zones, 2019–2020.

Sugar	Norte Grande	Norte Chico	Zona Central	Zona Sur	Zona Austral
Monthly intake average (days)	10.5 ± 11.1	9.4 ± 10	10.5 ± 11.3	6.8 ± 8.7	6.3 ± 8.1
Highest (days)	30	30	30	30	30
n (%) [Households]	36 (20.2)	43 (14.4)	363 (21.8)	169 (9.4)	7 (7.1)
Lowest (days)	0	0	0	0	0
n (%) [Households]	15 (8.4)	18 (6)	160 (9.6)	192 (10.1)	11 (11.2)
Mode (days)	2.1	2.1	2.1	2.1	2.1
n (%) [Households]	71 (40)	123 (41)	664 (40)	900 (50)	54 (55)
No. of households above recommended (%) *	88 (49.4)	149 (50)	840 (49.5)	593 (33.4)	38 (38.8)

* Details of communes with more than 50% of households above recommended intake can be found in Appendix A.

## Data Availability

The datasets used in this study were authorized for analysis under a privacy agreement with the Ministry of Social Development and Family of Chile.

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
