# Peer review of "Using Household Dietary Diversity Score and Spatial Analysis to Inform Food Governance in Chile"

_nutrients, 2024, doi:10.3390/nu16172937_

Round 1

Reviewer 1 Report

Comments and Suggestions for Authors

Martín del Valle and colleagues present a study to explore how HDDS and spatial visualization can inform food governance in Chile.

The use of a national household database from a government program lends credibility and relevance to the findings.

While the abstract mentions the geographical impact on food consumption patterns, it could be more specific. How exactly do these regions differ in terms of access to food, cultural practices, or economic conditions? Including a brief explanation would make the findings more understandable.

The conclusions could be more specific. Instead of broadly stating their importance, the suggests a few concrete ways the described tools could be used to enhance food security and governance strategies in Chile.

I believe that introducing more 2 or 3 related references would benefit the part of the discussion.

Author Response

  • Specify regional differences in the abstract:

    • Due to the word limit in the abstract, regional differences were added as two new columns indicating climate and agricultural production types in Table 2, highlighted in green.
  • Make the conclusions more specific and practical:

    • The following practical recommendation was added: "The practical recommendations based on the results and analysis of this study are as follows: (1) When selecting households that could participate in the program, it is essential to analyze their demographic characteristics to ensure that the inclusion of these households is based on identifying social conditions that make them more vulnerable to having lower food diversity. (2) Understanding the distribution of consumption of critical food groups across Chilean territory can also aid in the prior planning of the logistics required to improve access to these foods in the participating households,” and it is highlighted in green.
  • Add additional references in the discussion:

    • References regarding climate types in Chile and agriculture by macro-region (Sarricolea et al., 2017 and Meza et al., 2021) were added to the discussion and highlighted in green.

Reviewer 2 Report

Comments and Suggestions for Authors

Dear authors 

I want to congratulate the authors on their insightful work addressing the critical issue of food security in Chile. The impressive work in analyzing a substantial dataset comprising 4,407 households demonstrates the authors' dedication and expertise. Thoroughly analyzing such a large dataset is no small feat. 

The manuscript presents an analysis of food security in Chile, focusing on the Household Dietary Diversity Score (HDDS) and spatial visualization techniques. The study utilizes a substantial national household database (n=4,407) from the Family Support Program of Food Self-Sufficiency (FSPFS) to explore socio-demographic and geographical determinants affecting food consumption patterns among vulnerable populations. The findings highlight significant dietary deficiencies and the need for targeted interventions.

General Comments

The manuscript addresses a critical issue of food insecurity in Chile, particularly among vulnerable populations. The use of a large dataset enhances the reliability of the findings. However, several areas of the manuscript could be improved for clarity, depth, and overall impact.

Specific Comments

The title is informative but could be more concise. Consider simplifying it to enhance clarity.

The abstract effectively summarizes the study 

The introduction provides a good background 

The methodology section is generally well-structured

The results are presented clearly, with specific data and statistics that illustrate dietary intake patterns among vulnerable households in Chile. The use of mean and median values for the Household Dietary Diversity Score (HDDS) and the frequency of food group consumption provides a solid foundation for understanding the dietary diversity and food security issues these households face.

In Table 9, a description of the symbols used should be added.  Include a footnote at the end of the table. 

The discussion section addresses important implications. However, the conclusions should reflect the research's specific aims and summarize how the findings address those objectives. The conclusions presented contain elements that could be more appropriately integrated into the discussion. By incorporating these elements into the discussion, the authors can comprehensively analyze their findings, contextualize them within the existing literature, and offer actionable recommendations for policy and future research. 

References are up-to-date and are relevant to the study.

Overall Assessment

The manuscript presents valuable insights into food security in Chile, utilizing a robust dataset to inform food governance. With revisions addressing the comments and suggestions outlined above, the manuscript has the potential to make a significant contribution to the field. Moreover, it is recommended that the authors thoroughly review the English language used. This includes checking for clarity, grammatical accuracy, and consistency in terminology.

Comments on the Quality of English Language

It is recommended that the authors thoroughly review the English language used. This includes checking for clarity, grammatical accuracy, and consistency in terminology.

Author Response

  1. Review the English language for clarity and precision:

    • The language was reviewed for clarity and precision.
  2. Simplify the title:

    • The title was simplified and highlighted in green.
  3. Include the footnote in Table 9:

    • The footnote was added to Table 9 and highlighted in green.
  4. Restructure the conclusions, moving content to the discussion:

    • Content related to the importance of considering that nearly 90% of households were led by women (though no significant relationship with HDDS was found) was moved to the discussion. Practical considerations were also added to the conclusions.

Reviewer 3 Report

Comments and Suggestions for Authors

The paper “Enhancing Food Security through Data Insights: Exploring how Household Dietary Diversity Score and Spatial Visualization in Chilean vulnerable population can inform food governance” contributes to the growth of literature for research on the area on the nutrition of the population and food policy.

However, before the manuscript acceptations for publication in “Nutrients” the following items should be revised:

The studies concern the main groups of food products. Therefore, I suggest introducing the sentence "Basic research" to the title of this article.

Methods

What was the method of selecting households (income or other?

Did the authors ask about the types of products from a given group? For example, types of fat (saturated, unsaturated).

It is also applied to other products, e.g. dairy or fruit.  Quantity does not mean quality.

What statistical analysis was used? The authors did not describe this part in the methodological part, so what software was used?

Results

I suggest deleting Table 3 and listing the product groups in points.

Figure 5 is not very visible, similarly 6, 7, 8 and 9

I suggest adding "in the years ......." to the titles of tables and graphs

I suggest

When describing the results (e.g., fat or protein consumption), paying attention to that quantity does not mean quality.

Author Response

  • Change the title to include “Basic research”:

    • The title was simplified as suggested by Reviewer 2, avoiding the use of "basic research."
  • Describe statistical analyses and software used:

    • Information on multiple linear regression, the “RStudio” software, and the packages used were included in the “Materials and Methods” section of the manuscript.
  • Include in the methodology how households were selected:

    • The methodology now includes: “The households invited to participate in this program are those belonging to the 40% most vulnerable population in Chile, according to the socio-economic characterization survey conducted nationwide,” highlighted in yellow.
  • Add details about products and quality vs. quantity:

    • The methodology notes that the public program’s database does not ask about the type of product by food group, so details on quality versus quantity were not available.
  • Update reference numbers:

    • Reference numbers have been updated.
  • Add periods to table and figure titles:

    • Periods were added to the titles in the abstract, results, tables, and figures. This is highlighted in yellow in the abstract and results.
  • Improve the visibility of Figures 5 to 9:

    • The attached supplementary material file with figures has been provided in higher quality.
  • Eliminate Table 3 and list products in bullet points:

    • Table 3 was removed and food groups were listed in bullet points. These changes are highlighted in yellow.
